# Classification of multiparametric correlation MRI signals using deep neural networks

**Sebastian Endt**[1,2,3]                                   SEBASTIAN.ENDT@THI.DE

**Tobias Lachermeier**[1]                                   TOL8455@POSTEO.DE

**Marion I. Menzel**[1,2,3]                                 MARION.MENZEL@THI.DE

[1] *AImotion Bavaria, Technische Hochschule Ingolstadt, Ingolstadt, Germany*
[2] *Technical University of Munich, Garching, Germany*
[3] *GE HealthCare, Munich, Germany*

**Editors:** Accepted for publication at MIDL 2024

## Abstract

Correlation MRI is a promising microstructure imaging technique, but its reconstruction remains highly ill-conditioned. We propose to first classify correlation signals, and achieve high accuracy for classification into single- vs. multi-component. Further we establish a solid baseline for predicting the exact number of sub-compartments.

**Keywords:** Magnetic Resonance Imaging, MRI, multiparametric correlation imaging, multi-component relaxometry, classification, microstructure

## 1. Introduction

Multiparametric correlation magnetic resonance imaging (MRI) and multi-component relaxometry are promising techniques for the quantification of soft tissue microstructure. However, the need for many different contrasts collides with limited scan times, especially in clinical exams. This leaves the reconstruction of tissue sub-compartments in the form of multiparametric spectra as a highly ill-conditioned inverse problem that requires strong regularization (Canales-Rodríguez et al., 2021; Benjamini and Basser, 2020).

We believe that prior knowledge will allow for novel strategies in the quantification of tissue sub-compartments and propose the classification of correlation signals using neural networks. We aim to simultaneously answer two questions:

- Are there multiple sub-compartments present?

- How many sub-compartments are there?

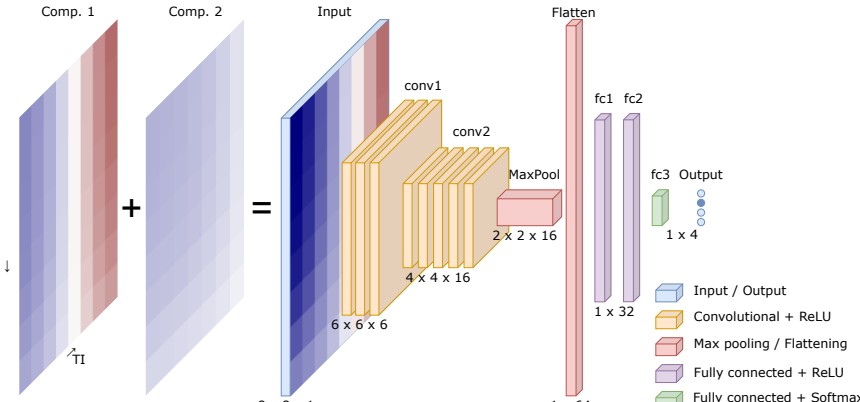

Figure 1: A weighted sum of compartmental 2D signals is fed into our CNN. The network comprises two convolutional layers, followed by three fully-connected layers. We distinguish four output classes, representing the number of sub-compartments $n_c$ (here: $n_c = 2$). Adapted from (Leung, 2021).

## 2. Methods

### 2.1. Data set

The data for our deep learning approach consists of multi-contrast input signals and the corresponding class, i.e. number of sub-compartments $n_c$. We consider the four cases $n_c \in \{1, \ldots, 4\}$, each contributing 25 % of the data. In total, $10^6$ samples are generated.

To simulate tissues, we choose $n_c$ random T1$\in \{50 : 50 : 5000\}$ms, random T2$\in \{5 : 5 : 500\}$ms, and signal fractions $f \in [0, 1]$, with $f \geq 0.05$ and $\sum_{i=1}^{n_c} f_i = 1$. We assume a set of $8 \times 8 = 64$ unique combinations of scan parameters TI and TE, taken from an actual correlation imaging experiment (Endt et al., 2023). Given the set of $n_c$ tissue parameters T1, T2, and $f$ and the set of 64 different scan parameters $\mathbf{TI}$ and $\mathbf{TE}$, the multiexponential signal $\mathbf{M}$ of size $8 \times 8$ is computed as follows:

$$\mathbf{M} = \sum_{i=1}^{n_c} f_i \cdot \exp\left(-\frac{\mathbf{TE}}{\text{T2}_i}\right) \cdot \left(1 - 2 \cdot \exp\left(-\frac{\mathbf{TI}}{\text{T1}_i}\right)\right). \tag{1}$$

Random Gaussian noise between 1 % and 5 % is added to the multiexponential signals. Lastly, all signals are normalized to their 2-norm $\|\mathbf{M}\|_2 \overset{!}{=} 1$.

The simulated data is split into training, validation and test data according to the ratio 70 %-20 %-10 %, while both varying number of sub-compartments $n_c$ and different noise levels are equally present in all three data sets.

### 2.2. Architecture and learning strategy

We use a convolutional neural network (CNN) with two ReLU-activated convolutional layers with a $3 \times 3$ kernel and 6 and 16 channels, respectively. The second convolution is followed by a $2 \times 2$ max-pooling layer and 20 % dropout. Then we add two ReLU-activated fully

connected layers with 32 nodes and 20 % dropout each, and a softmax-activated output layer with four classes. The architecture is shown in Fig. 1.

The network is trained on training data, minimizing the cross-entropy between outputs and reference labels using stochastic gradient descent (SGD) with a momentum of 0.9. The learning rate decreases step wise from 1 to 0.01 over a total of 3000 epochs. We use validation data to choose the final model, which is finally tested on previously unseen test data. The network is trained in Python 3.10 using PyTorch 2.2 (Paszke et al., 2019). Results are evaluated using scikit-learn 1.3 (Pedregosa et al., 2011). The code is publicly available on https://github.com/SebastianEndtTHI/CorrelationClassification.

## 3. Results

The binary classification task distinguishing between one ($n_c = 1$) or more ($n_c \geq 2$) compartments reached an accuracy of 94.1 % (precision 0.88, recall 0.88), and a Receiver Operating Characteristic (ROC) area under the curve (AUC) of 0.98. The confusion matrix is shown in Fig. 2(a). The multi-class prediction of the number of compartments reaches an accuracy of 61.2 % (precision 0.60, recall 0.61) and is further evaluated using a Confusion Matrix and ROC curves, as shown in Fig. 2(b) and 2(c), respectively, detailing class specific results.

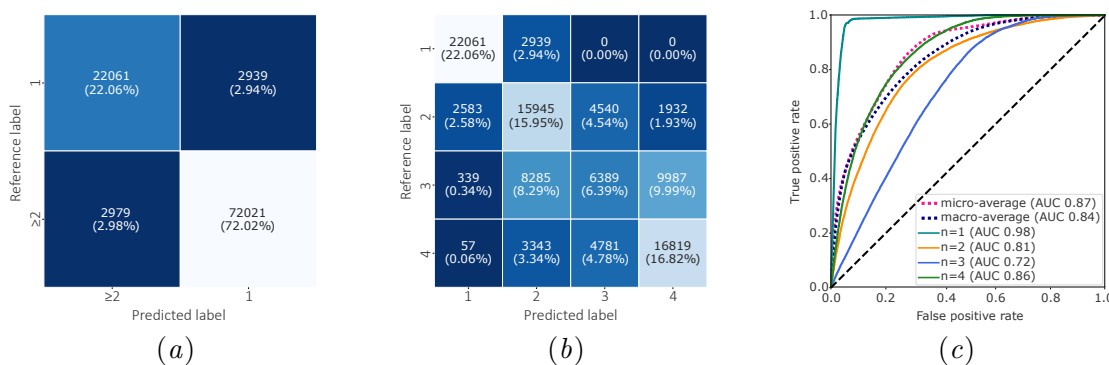

$(a)$ $\qquad$ $(b)$ $\qquad$ $(c)$

Figure 2: (a) Binary confusion matrix for the prediction whether there are multiple compartments. (b) Multi-class confusion matrix for the estimation of the number of compartments $n_c$. (c) ROC curves for the multi-class problem, including individual one-vs-rest curves and averages. $n = 1$ is equivalent to the binary task.

## 4. Discussion

The binary classification tasks of distinguishing signals with multiple compartments against signals with one single compartment reached a high accuracy and may soon be used as a prior to stabilize both constrained optimization algorithms and recently emerging deep learning approaches for the reconstruction of spectra (Yu et al., 2021; Endt et al., 2021).

For the multi-class classification task of predicting the number of compartments $n_c$, we reach an accuracy of 61.2 %. This establishes a solid baseline for this novel problem, which is subject to further development of the methodology in the future, i.e. by making use of spatial context to stabilize the classification of a whole image slice or volume.

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
