# OpenReview forum: "Classification of multiparametric correlation MRI signals using deep neural networks"
_MIDL.io/2024/Short_Papers — MIDL 2024 Short Papers_

### Official Review · Reviewer_mgiW · 2024-04-24

**Confidence:** 3
**Final Rating:** 4

**Review:**

This paper focuses on the classification multiparametric correlation MRI signals. Correlation MRI is a promising technique for the quantification of soft tissue structure but its reconstruction is ill-conditioned.

Strengths:
- code publicly available
- Well-written short paper.
- Clear application paper with potential value for researchers
- high accuracy for the binary classification

weakness:
- accuracy for the multi-class prediction of the number of compartments is  61.2%

---

### Decision · Program_Chairs · 2024-04-26

Accept